# Molecular Pharming of the Recombinant Protein hEGF-hEGF Concatenated with Oleosin Using Transgenic *Arabidopsis*

**DOI:** 10.3390/genes11090959

**Published:** 2020-08-19

**Authors:** Weidong Qiang, Tingting Gao, Xinxin Lan, Jinnan Guo, Muhammad Noman, Yaying Li, Yongxin Guo, Jie Kong, Haiyan Li, Linna Du, Jing Yang

**Affiliations:** College of Life Science, Engineering Research Center of the Chinese Ministry of Education for Bioreactor and Pharmaceutical Development, Jilin Agricultural University, Changchun 130118, China; qiangweidong@jlau.edu.cn (W.Q.); gaotingting@jlau.edu.cn (T.G.); lanxinxin@jlau.edu.cn (X.L.); guojinnan@jlau.edu.cn (J.G.); noman@jlau.edu.cn (M.N.); liyaying@jlau.edu.cn (Y.L.); guoyongxin@jlau.edu.cn (Y.G.); kongjie@jlau.edu.cn (J.K.); hyli99@163.com (H.L.); linnad@jlau.edu.cn (L.D.)

**Keywords:** epidermal growth factor, oil body, transdermal absorption, proliferation activity

## Abstract

We set out to assess the NIH/3T3 cell proliferation activity of Arabidopsis oil body-expressed recombinant oleosin–hEGF–hEGF protein. Normally, human epidermal growth factor (hEGF) is purified through complex process, however, oleosin fusion technology provides an inexpensive and scalable platform for its purification. Under a phaseolin promoter, we concatenated oleosin gene to double hEGF (hEGF–hEGF) with plant-preferred codons in the expression vectors and the construct was transformed into *Arabidopsis thaliana* (Arabidopsis). The transgenic Arabidopsis was validated by RT–PCR and the content of recombinant protein oleosin–hEGF–hEGF was quantified by western blot. Subsequently, the proliferation assay and transdermal absorption were determined by MTT method and immunohistochemical staining, respectively. First, the expression level of hEGF was recorded to be 14.83-ng/μL oil body and due to smaller size transgenic oil bodies expressing the recombinant oleosin–hEGF–hEGF, they were more skin permeable than those of control. Second, via the staining intensity of transgenic oil bodies was greater than EGF at all time points via immunohistochemical staining in transdermal absorption process. Lastly, activity assays of oil bodies expressed oleosin–hEGF–hEGF indicated that they stimulated the NIH/3T3 cell proliferation activity. Our results revealed oil-body-expressed oleosin–hEGF–hEGF was potential new material having implications in the field of medicine.

## 1. Introduction

Human epidermal growth factor (hEGF) is a small molecule polypeptide consisting of 53 amino acids. It contains an hEGF-domain which is conserved by a six-cysteine residue motif. hEGF plays a key component of complex involved in cell differentiation, proliferation and migration [1]. It is an effective mitogen in vitro and in vivo, not only for epithelial cells, but also for mesenchymal and endothelial cells. hEGF mediates an array of pathologic and physiological processes such as regeneration, growth, embryogenesis and tissue repair [2]. In addition, the effect of hEGF on inflammatory responses to burn injuries was examined by studying the extent of neutrophilic leukostasis. hEGF has a significant role in promoting the healing of wounds and ulcers; therefore it is widely used in the medicine and cosmetics fields [3,4].

The oil body is a spherical small organelle with a diameter of 500–2500 nm, in which the plant seed oil is stored [5]. The oil body is a subcellular particle of plant seed storage, which is the smallest organelle storage lipid in plants, mainly TAG, which provides energy for subsequent life activities and active metabolic processes [5,6]. Oleosin proteins of different plant origins have the same structural characteristics and all have three basic domains; (1) the N-terminal amphiphilic region (both hydrophilic and lipophilic), (2) intermediate highly hydrophobic region and (3) the variable region of 33 to 40 amino acids at the C-terminus [7]. The C-terminus and N-terminus exposed to the cytoplasm are implanted on the oil-body surface. The N-terminal amino acid sequence is not conserved, and most of its residues are associated with the surface of the oil body towards the cytosol. The C-terminal region is distributed on the oil-facing side of the oil body [8]. Oleosin molecules could be interlinked with small peptides of specific amino acid residue [9]. The oleosin sequences exist in N-terminal or C-terminal for insertion of exogenous proteins. The method of recombinant exogenous protein production could be fused to N-terminal or C-terminal of oleosin anchored to the oil body surface in favor of purification.

Expressing the recombinant therapeutic proteins in oil body systems is scalable, safe and cost-effective. Oil body may carry an effective constituent to be transported to host [9]. Oil body emulsions can be used in a wide variety of applications, such as in food and feed, pharmaceuticals, personal care products and industrial products. Oil body emulsions of therapeutic hormonal peptide-fused recombinant oleosin have been produced and applied to the population as products for improving fish food and manufacturing personal care [9]. The formulation of personal care products is also labeled with this liposome oil body [10].

Currently, the hEGF protein is produced via bacterial expression systems, but their purification was a difficult and complicated procedure that demanded several steps and often resulted in very low recovery of the protein [1]. However, the exogenous protein expressed by the oil bodies of the plant seed was convenient and simple to purify. The purification step is reduced lowering the cost. Oil body can be easily separated from seeds by centrifugation [9,11,12]. In general, hEGF is usually difficult to express and purify in prokaryotic systems because it has a small molecular mass and thus the exogenous recombinant proteins can be produced cost-effectively in plant system [9,13].

In this study, we fused hEGF to N-terminus of oleosin and recruited plant oil body to express the recombinant oleosin–hEGF–hEGF, hence acting as a bioreactor. The recombinant protein expression level up to 14.83 ng/μL was obtained in transgenic Arabidopsis seeds. Through staining and activity assays, we demonstrated that the transgenic oil bodies were smaller and easier to permeate into skin stimulating NIH/3T3 cell proliferation. Western blot have also confirmed that transgenic oil bodies can activate EGFR.

## 2. Materials and Methods

### 2.1. Experimental Materials

Mature seeds of Columbia type Arabidopsis were kindly provided by the Jilin Agricultural University, China. The plasmid pOTB was reserved from Jilin Agricultural University, China. The pOTB plasmid contained the Arabidopsis *oleosin1* (*OLE1, AT4G25140*) gene (Gen Bank accession number X62353.1), phaseolin promoter/terminator (sequence information was provided by Professor Chao Jiang SemBioSys company), with 35S-bar gene (GenBank: AF218816.1) and nos terminator (GenBank accession number AF234297.1). Rabbit anti-hEGF polyclonal antibody was purchased from Abcam Co., Ltd. (Abcam, Cambridge, MA, USA) and goat anti-rabbit IgG/AP from Promega (Madison, WI, USA). Dulbecco’s modified Eagle’s medium (DMEM) was purchased from Invitrogen company (Carlsbad, CA, USA). Methyl thiazol tetrazolium (MTT) was bought from Gold Biotechnology (St. Louis, MO, USA). Twenty-four male and female ICR mice were weighed 20–22 g (Yisi experimental animal technology company China).

### 2.2. hEGF Gene Cloning and Vector Construction

*hEGF* nucleotide sequences (Gen Bank accession nos. JQ346088.1) were optimized as the codon usage table Arabidopsis (www.kazusa.or.jp/codon/). The pUC–hEGF–hEGF vector was constructed by Suzhou Jinweizhi Biological Technology Co., Ltd. (Suzhou, China). The T-DNA of the pOTB vector included a phaseolin promoter/terminator, the fusion gene, the *35S-Bar* gene and the nos terminator. The pOTB plasmid was extracted from Escherichia coli and digested with *Nco*I and *Hin*dIII. The *hEGF–hEGF* gene was inserted into the pOTB plasmid. The new recombinant plasmid was named pOTB–hEGF–hEGF. A fragment of 2646 bp consisting of the phaseolin promoter and oleosin fused to plant-optimized double *hEGF* gene was created (Figure 1). The recombinant plasmid from DH5α was verified by PCR and restriction digestion. The recombinant plasmid pOTB–hEGF–hEGF was subcloned into *Agrobacterium tumefaciens* strain EHA105 competent for the stable transformation of Arabidopsis.

### 2.3. Transformation of Arabidopsis

EHA105 cells were centrifuged for 15 min at 4000 rpm, the bacteria was collected and resuspended in MS medium at OD 0.8–1.0. The mature seeds were kept at 4 °C for 3 d, then the seeds were sowed in soil with a pipette and cultured for 2 d in the dark. After the seeds germinated, they were cultured under weak light. The plantlets were further cultured under fluorescent lamps after four leaves grown [14]. After 40 d culture, the plantlets were infected in resuspended MS medium by floral dip method. The infected plants were placed flat in the dark overnight and were then harvested for T1 seeds. The T1 generation of the transgenic seeds were obtained, they were reseeded and sown at 25 °C under 16 h of light. When the plants had 4–6 leaves, the plants were selectively transformed with 0.5% glufosinate which was sprayed every other day for a total of three times. Surviving T2 transgenic plants were transplanted and cultured in pots from which T3 seeds were collected.

### 2.4. Extraction of Oil Bodies

Oil bodies were extracted following the protocol [15]. About 20 mg of Arabidopsis seeds were weighed for screening. They were mixed with 200 μL of sodium phosphate buffer (PBS, pH 7.5) and disrupted to a particle-free state by a mortar. Then, the mixture was spun at 12,000× *g* and 4 °C for 15 min to obtain a pure portion [16,17]. The pellet and liquid were discarded, and the oil body was resuspended in 200 μL of PBS. This step was repeated three times. Finally, Arabidopsis pure oil bodies were obtained.

### 2.5. Validation of Transgenic Arabidopsis Expressing Recombinant Oleosin–-hEGF–hEGF

Total RNA was extracted from 20 mg of Arabidopsis seeds by Trizol. One μg of total RNA was used to make cDNA followed by RT–PCR amplification. The RT–PCR program included a pre-denaturation of 94 °C for 8 min followed by 30 cycles of 94 °C for 30 s, annealing step of 56 °C for 30 s and an elongation step of 72 °C for 1 min. Amplified bands of the product were observed on a 1.2% *w/v* agarose gel.

### 2.6. Western Blot Analysis of the Oil Body-Expressed Oleosin-hEGF–hEGF

For western blot analysis, oil bodies from Arabidopsis were prepared with extraction buffer at a ratio of 50 μL buffer per 5 μg seeds and separated by SDS-PAGE. The loading quantity of sample of SDS-PAGE was 10 μg total protein. The proteins were electrotransferred onto 0.45 μm polyvinylidene difluoride membranes (PVDF). For immunodetection, the PVDF membrane was incubated with a polyclonal rabbit anti-hEGF antibody and washed four times with TBST. Then the secondary antibody (goat anti-rabbit IgG/AP antibody) was added to PVDF membrane. Immunoreactive bands were visualized with AP coloration reagents. The sampling amount of the oil body was 10 µg in western blot. We analyzed the data of fusion protein accumulation in the seeds using quantity one software.

### 2.7. Measuring of the Oil Body Particle Size

The particle diameter of transgenic oil bodies was determined using laser particle analyzer. The transgenic oil bodies were diluted using PBS buffer, where they dispersed into individual particles. They were measured by a laser light scattering instrument (Mastersizer 2000). The tests were repeated three times. All data were statistically calculated using GraphPad Prism 6.01 software.

### 2.8. Microstructure Detection of the Oil Body

The transgenic oil bodies were colored and photographed using dyestuff of Nile red in microscope. The transgenic oil bodies were treated using PBS buffer solution which were uniformly dispersed to ensure their homogeneity. The transgenic oil body suspensions were dyed and stewed in dark. The microscopic dynamic state was observed and photographed at the magnification of 10 × 40 under the fluorescence microscope.

### 2.9. Transdermal Absorption of the Transgenic Oil Body

Twenty-four male and female ICR mice (Yisi experimental animal technology company, China) were divided into 4 groups randomly (*n*−6); saline group (NS), hEGF group (positive control group), wild type oil body group (negative control group) and oil-body-expressed oleosin–hEGF–hEGF group (*n* = 10). These mice were injected with 5% chloral hydrate solution in vivo. The hair of their backside were depilated. The blank control group was 40 μL of PBS used on the back. The positive control was applied to the back with 40 μL of PBS solution containing 20 μg of EGF protein, compared to the treated group. The 40 µL of the oil body-expressed oleosin–hEGF- hEGF fusion protein was smeared as the sample group. Samples were smeared in 0.1 cm^2^ area of backside. The skin tissues were sampled via dosing for 15, 30 and 45 min. A part of these skin samples were made into paraffin sections. Tissue slices were stained with DAB kit and analyzed by immunohistochemical experiment. Another part of the skin was used for protein extraction using a whole protein extraction kit. The total protein of the extracted skin tissue was quantified by BCA and the total protein concentration were 6.9223 μg/μL, 7.2219 μg/μL, 7.1521 μg/μL (y = 0.9897x − 0.0056, R^2^ = 0.9998) and then samples were prepared for SDS-PAGE electrophoresis, the amount of protein loaded on each channel was uniformly 60 μg. Then the protein on the gel was transferred to the PVDF membrane and the primary antibody (polyclonal rabbit anti-*p*-EGFR antibody) and the second antibody (goat anti-rabbit IgG/AP antibody) were incubated to develop the color to detect the activation of *p*-EGFR by oleosin–hEGF.

### 2.10. Resistant to Proteolysis of Oil body-Expressed Oleosin-hEGF–hEGF

We took 10 μg each of oleosin–hEGF and hEGF protein in 25 mM ammonium bicarbonate solution, added 10 μL of 0.25% protease (trypsin) to 100 μL solution and incubated at 37 °C, according to solution volume: protease volume = 100:1. The hydrolyzed protein was taken out at regular intervals (0, 10, 15, 30, 45, 60, 90 min), an appropriate amount of 5× Loading buffer was added and then boiled in boiling water for 10 min to fully denature the protein and was then stored at −20 ℃ for later use. Oil body-expressed oleosin–hEGF–hEGF and hEGF standard protein were digested with protease and run on SDS-PAGE electrophoresis. After SDS-PAGE, Coomassie blue staining was performed, and the integrity of the protein structure was observed after decolorization.

### 2.11. Proliferation Assay of the Oil body-Expressed Oleosin–hEGF–hEGF

A 20 μg of the T3 generation transgenic seeds were immersed in 100 μL of PBS (pH 7.5) and then thoroughly ground. The oil bodies were extracted by gradient centrifugation. The above oil bodies were mixed with DMEM to examine the effect of promoting proliferation of NIH/3T3 cells. First, NIH/3T3 cells were cultured in DMEM low-sugar medium. Then they were grown in 96-well plates at a density of 5 × 10^4^ cells/mL. Cells were treated with different samples for 48 h. After 2 d incubation with different samples, 25 μL of 5 mg/mL MTT solution was added to each well. The cells were incubated at 37 °C for 4 h and 100 μLof dimethyl sulfoxide was added to the culture solution. The absorbance was measured at 570/630 nm in a microplate reader model 450 nm.

### 2.12. Statistical Analysis

The samples of each experiment were carried out in 3 biological replicates. The data of repeated experiments are expressed as the mean and standard deviation (mean ± SD). For differences among groups, we used one-way ANOVA (analysis of variance) and with *p*-value < 0.05, the difference was considered significant.

## 3. Results

### 3.1. Transformation and Detection of Arabidopsis

T-DNA region connecting to the *AtOLE-hEGF–hEGF* gene was constructed (Figure 1A). The recombinant plasmids pOTB–hEGF–hEGF were detected by PCR (Figure 1B). To get T3 Arabidopsis lines, we first carried out Basta screening of T1 plants during which most of the plants turned yellow and died. Basta-survived green plants were tested through PCR and only transgenic plants were allowed to grow. T2 seeds were harvested from transgenic plants and sowed again to get T3 seeds, which were proceeded for further experimentation. Thus, six lines of T3 homozygous seeds were obtained after dual screening (BASTA and RT–PCR). The separation ratio was 3:1 in line with Mendel rule. A portion of transformed seeds were detected for *hEGF* transcripts by RT–PCR. The analysis result showed that hEGF transcripts accumulated in transgenic seeds and the positions of the target bands were according to the expected ones. The bands of four transgenic lines for the plant-preferred double *hEGF* gene appeared at 327 bp (Figure 1C). We constructed a gene expression cassette containing the oleosin–hEGF–hEGF gene under the control of the phaseolin specific promoter (phaP)/terminator (phaT). The phaseolin promoter controls tissue-specific expression during seed development. This construct enabled the fusion gene to be specifically expressed in Arabidopsis seeds. The oleosin gene acted as a signal peptide to carry hEGF anchoring on oil body surface and express in Arabidopsis seeds. The oleosin may interact to form the exogenous protein and the oil bodies containing such exogenous proteins could be extracted. hEGF protein does not need to be separated from oil bodies.

### 3.2. Expression Analysis of Fusion Protein in Transgenic Arabidopsis

To detect the specificity of fusion protein, oil bodies were separated and purified from the wild type seeds and T3 transgenic seeds. We performed semiquantitative accumulation of the recombinant protein in T3 transgenic Arabidopsis seeds by western blot. First, various concentrations of hEGF standard protein were tested by western blot (Figure 2A) and the grayscale value was analyzed as shown in Figure 2B. Afterwards, the standard curve was drawn (Figure 2C). The X-axis shows various concentrations of hEGF standard protein, while the Y-axis shows the gray value. When the quantity of hEGF standard protein was 1000, the gray value was 7220, whereas; the regression equation was: y = 808.57x + 2120, R^2^ = 0.9902 (Figure 2C). Meanwhile, the fusion protein in T3 Arabidopsis seeds (Figure 2E) and gray value of each was band were analyzed with Quantity One software. Next, the expression level of the target protein was determined according to the grayscale value of the target band. Finally, the expression level of the target protein was analyzed according to the grayscale value of the target protein band, such as the highest expression level of target protein was calculated as 14.83 ng/µL oil body as shown in Figure 2F.

### 3.3. Analysis of Particle Size and Microstructure of Transgenic Oil Body

Particle size distribution of the oil body was determined by laser light scattering instrument. The particle size of the oil body to measure was for the sake of strong absorption capacity. The smaller the particle size, the easier to infiltrate into skin [18]. The average particle size of transgenic oil bodies was 1227 nm and the average particle size of normal oil bodies was 1556 nm as shown in Figure 3A. The average particle size of transgenic oil bodies was smaller than normal oil bodies. The oil bodies were stained red via Nile red dyestuff and observed under fluorescence microscope. As shown in Figure 3B, the overall diameter of transgenic oil bodies was smaller than normal oil bodies. The oil body is a subcellular organelle of the stored oil in the plant seed, which is a small sphere with a diameter of 500–2500 nm. The transgenic oil bodies were 1000–1400 nm in diameter while the wild type were 1400–1600 nm in diameter. The particle size was reduced in transgenic Arabidopsis which was ideal for skin absorption.

### 3.4. Stability Analysis of Oil Bodies-Expressed Oleosin–hEGF–hEGF

The stability between hEGF protein and oleosin–hEGF–hEGF expressed by oil bodies was compared. After oleosin–hEGF–hEGF and hEGF proteins were treated with protease, SDS-PAGE analysis was performed on the proteins at different processing times. The results are shown in Figure 4A. It can be seen that after protease digestion for 10 min, hEGF (6.5 KDa) obviously begins to degrade and its band depth weakens. By the 45th minute, hEGF was almost completely hydrolyzed, and no obvious band was observed (Note: proteins with larger bands are heteroproteins). However, oleosin–hEGF–hEGF (31.5 KDa) did not degrade significantly during the entire digestion process, as shown in Figure 4B. During the 90-min hydrolysis process, there was no significant change in the band position and band intensity, indicating that the fusion protein was not protease hydrolysis has a certain resistance to hydrolysis, which improves the stability of protein.

### 3.5. Transdermal Absorption of the Transgenic Oil Body

The capacity of transdermal absorption was compared between transgenic and normal oil body. hEGF protein as position control was smeared on the mice’s backside for 15, 30 and 45 min. The results showed that the absorption capacity of oleosin–hEGF–hEGF expressed by oil bodies was better than hEGF protein because some unabsorbed hEGF protein remained on the skin in this group during the 45-min transdermal absorption test. It was verified through immunohistochemical staining as shown in Figure 5. Compared with the same hEGF protein, the transgenic oil body group showed more positive cells, and at each time point, the staining intensity of oleosin–hEGF–hEGF expressed by oil bodies was greater than that of hEGF protein and normal saline treatment group. Biologic replicate test data are given is Appendix A. More absorbed protein could be concentrated in the subepidermal tissue and hair follicles.

### 3.6. Analysis of p-EGFR Receptor Activation

In order to find if oleosin–hEGF–hEGF could activate the EGFR receptor, we performed protein immunohybridization on skin total proteins at various time points in the transdermal absorption experiment. Figure 6 illustrates the results of *p*-EGFR receptor activation analysis. The recombinant protein oleosin–hEGF–hEGF (Figure 6A) and hEGF standard protein (6B) flaunted bands while normal saline (6C) and WT oil body (6D) were put as negative controls where bands are absent. With the transdermal absorption of oleosin–hEGF–hEGF and hEGF standard protein, the EGFR receptor began to activate at 0–15 min.

### 3.7. Oil Body-Expressed Fusion Protein Activity Analysis in NIH/3T3 cell

To assess the activity of transgenic oil bodies, we selected NIH/3T3 cell. The proliferation capacity of the transgenic oil body was detected in NIH/3T3 cell. As bFGF exhibits better activity to promote NIH/3T3 cells proliferation, the proliferation capacity was of oleosin–hEGF–hEGF and bFGF was compared [19]. The bFGF protein (12.5I U/mL) was diluted with 2-fold, which enhanced NIH/3T3 cells proliferation. The normal oil body displayed statistically insignificant proliferation capacity, however, the oleosin–hEGF–hEGF protein had a dose-dependent cell proliferative effect on the NIH/3T3 cells (Figure 7).

## 4. Discussion

In a number of studies, plant expression systems have been exploited to produce foreign active proteins [16]. Such a system shows huge advantages in cost efficiency, product quality and safety. Oil body also counts among the platforms for expressing exogenous proteins. The oil body system carries several advantages such as, the exogenous proteins are implanted in the oil body surface, alleviating target protein purification thus making it easier to obtain a pure product. Similarly, it could be directly applied to the skin surface. Therefore, the oil body expression system represents an ideal system for the production of therapeutic proteins. SemBioSys Biotech successfully expressed human insulin in Arabidopsis seeds through the plant expression vector pSBS4405. The safflower oil body was also successfully expressed and its commercial production standard was qualified [20]. Through oleosin fusion technology, oleosin–hFGF9 protein was expressed in Arabidopsis seeds and had biological activity that stimulates the proliferation of NIH/3T3 cells. Similarly, recombinant *aFGF* in Arabidopsis and recombinant *hbFGF* in Arabidopsis and rice seeds were successfully expressed. They have all been confirmed to possess obvious biological activity [17,20,21]. The process of purifying hEGF from *E. coli* cells is much more complicated than purifying proteins from oil bodies. In the oil body system, hEGF targets the C-terminus of oleosin, thus the purification process is easier and simple and does not require protein folding [15,17].

Plant bioreactors have the irreplaceable advantages of animal and microbial bioreactors: the homozygotes obtained after selfing of genetically modified plants can ensure the stable inheritance of new traits. Similarly, the expression product can undergo post-translational processing and protein glycosylation in plant cells, making the three-dimensional structure more natural, and the recombinant protein and the natural protein have nearly the same immunogenicity and biologic activity. Plant expression systems produce medicinal substances with low cost, wide sources and easy mass production. Compared with animal and microbial reaction systems, plants do not contain pathogens that harm human health. However, the ratio of exogenous recombinant proteins to the total plant biomass is usually very low, which increases the cost of separation and purification. Therefore, how to increase the expression of exogenous proteins in plants, reduce downstream production costs and simplify the purification process is a problem in plant reactor research. However, the use of plant seeds to express foreign proteins has attracted much attention, mainly because of the high protein content of plant seeds. During the seed maturation process, about 95% of the water is actively lost, the hydrolytic enzyme activity in the cell is greatly reduced, and the recombinant protein can be stored in the seeds for a long time without being degraded. The oleosin in the seed has the characteristics of both hydrophilic and lipophilic. It is embedded in the surface of the oil body in a membrane embedding manner in the plant seed. Oil-body protein is expressed at a high level in oil crop seeds and is easily separated. To date, various proteins such as hirudin and FGFs have been expressed in the oil-body bioreactor, and the expression level can reach commercial standards [11,16,22].

In our study, we constructed a plasmid expressing the recombinant protein oleosin–hEGF–hEGF in the oil bodies of Arabidopsis seeds with an expression level of 14.83-ng/μL oil body, and recombinant fusion proteins were seen highly active. The oil bodies enhanced the stability of the target protein because it has a structure similar to that of the liposome. The diameter of transgenic oil bodies were smaller than normal oil bodies favoring efficient transdermal absorption. The staining intensity of transgenic oil bodies were greater than hEGF at all time points via immunohistochemical staining which demonstrated the efficient activity of the recombinant protein. Therefore, this new drug delivery system (oil bodies) can rapidly infiltrate hEGF into the skin, which has a very good activity to promote cell proliferation.

## 5. Conclusions

We constructed a plasmid expressing the recombinant protein oleosin–hEGF–hEGF in the oil bodies of *Arabidopsis* seeds with an expression level was 14.83-ng/μL oil body. The diameter of transgenic oil bodies were smaller than normal oil bodies favoring efficient transdermal absorption. The staining intensity of transgenic oil bodies were greater than hEGF at all time points via immunohistochemical staining which demonstrated the efficient activity of the recombinant protein. Activity assays of oil bodies expressed oleosin–hEGF–hEGF indicated that they stimulated the NIH/3T3 cell proliferation activity along with the activation of expression of EGFR in the skin. The stability test showed that the oil body can protect hEGF activity. This study furthers the concept of the recombinant protein oleosin–hEGF–hEGF as a potential new material having implications in the field of medicine.

## Figures and Tables

**Figure 1 genes-11-00959-f001:**
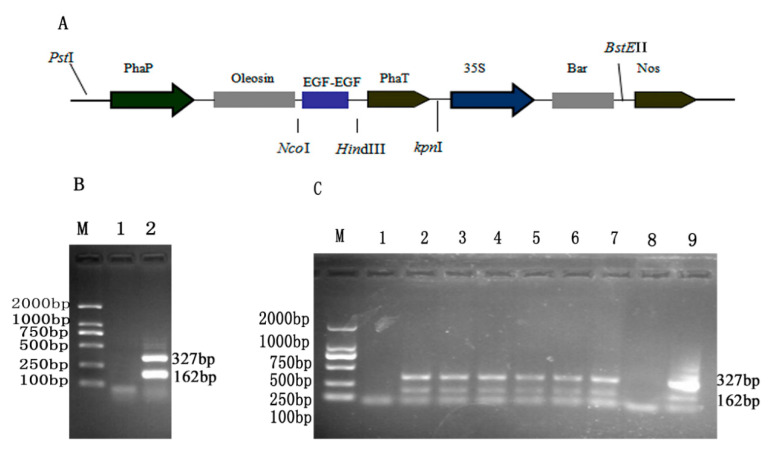
Construction and detection of pOTB–hEGF–hEGF. (**A**) Schematic diagram of T-DNA region from pOTB–hEGF–hEGF; (**B**) PCR detection of recombinant plasmid; (**C**) T3 transgenic Arabidopsis seeds were determined by RT–PCR; (M) DNA2000 marker; 1—negative control (wild type); 2–7—T3 transgenic seeds; 8—negative control (water); 9—positive control (pOTB–hEGF–hEGF plasmid).

**Figure 2 genes-11-00959-f002:**
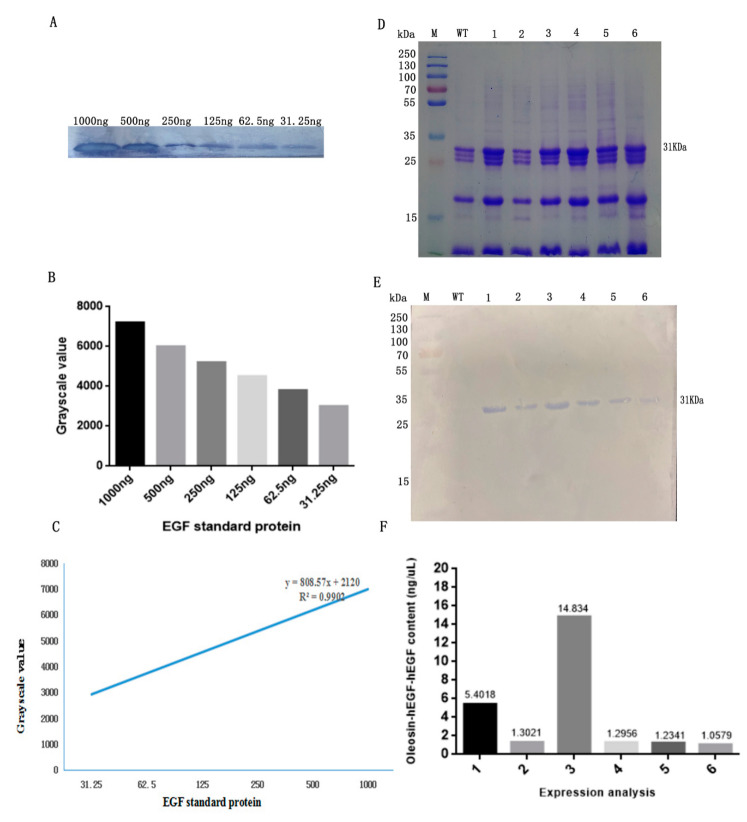
Protein expression from T3 transgenic Arabidopsis. (**A**) western blot; (**B**) grayscale values and (**C**) the standard curve of the grayscale values from various concentrations of hEGF standard protein; (**D**) SDS-PAGE of the oil body; (**E**) western blot of fusion protein and (**F**) quantitation of fusion protein from T3 Arabidopsis seeds. M—protein marker; 1–6—oil body in T3 Arabidopsis; WT—oil bodies from WT Arabidopsis.

**Figure 3 genes-11-00959-f003:**
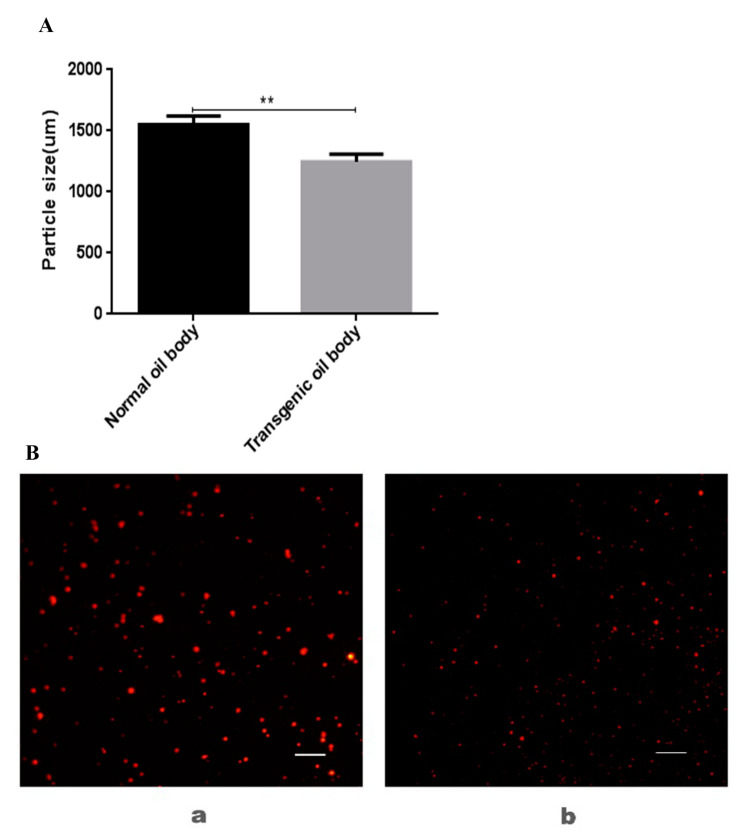
Measurement and observation of size and microstructure from transgenic oil body and normal oil body. (**A**) Particle size; (**B**) microstructure observation; (a) dyed normal oil bodies via Nile red; (b) dyed transgenic oil bodies via Nile red. The images were shown at 40 × 10 magnification (scale bars = 10 µm). ** *p* < 0.01 vs. normal group.

**Figure 4 genes-11-00959-f004:**
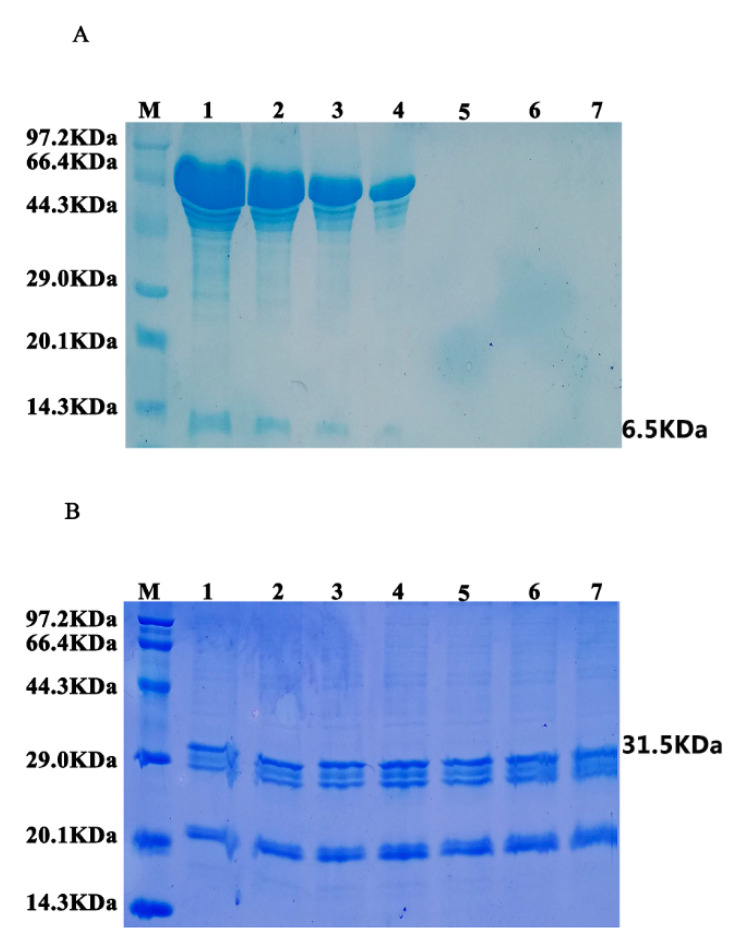
Analysis of hEGF protein and oleosin–hEGF–hEGF activation stability. (**A**) SDS-PAGE analysis of hEGF standard protein digested with protease at different times; (**B**) SDS-PAGE analysis of oleosin–hEGF–hEGF digested with protease at different times; (M) protein marker, 1–7—hydrolyzed protein of 0, 10, 15, 30, 45, 60, 90 min.

**Figure 5 genes-11-00959-f005:**
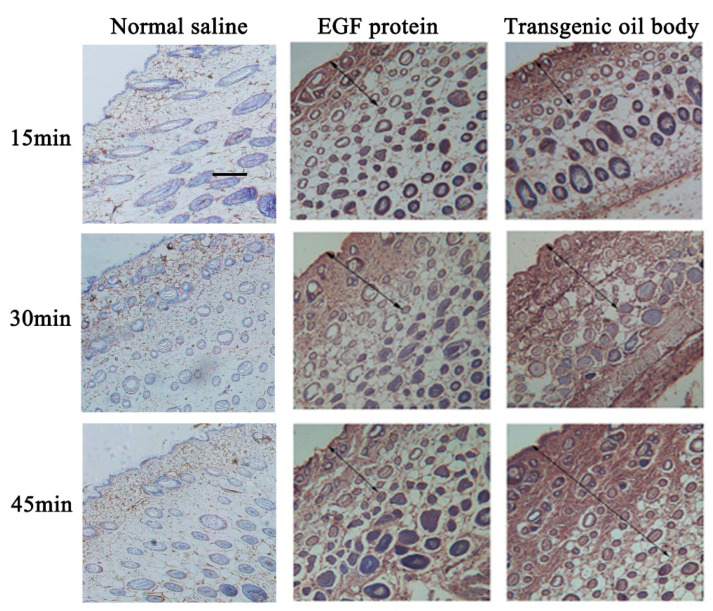
Ability of transgenic oil bodies bound hEGF to infiltrate into skin. The immunohistochemistry analysis of drug absorption at different times is shown. hEGF protein is the positive group and saline is the negative control group. Images shown at 40 × 10 magnification (scale bars = 50 µm).

**Figure 6 genes-11-00959-f006:**
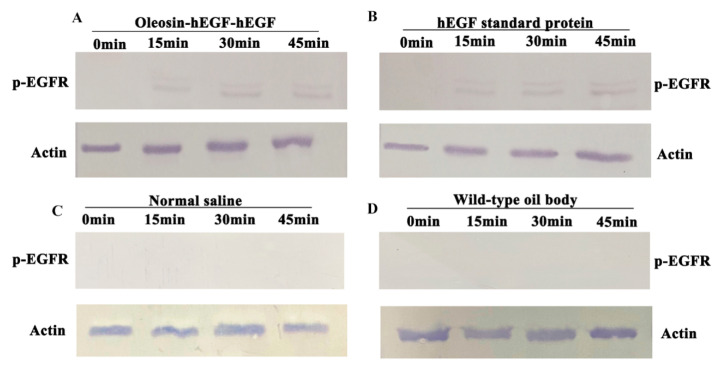
Analysis of activation of *p*-EGFR. Activation of *p*-EGFR after (**A**) oleosin–hEGF–hEGF; (**B**) hEGF standard protein; (**C**) normal saline and (**D**) WT oil body treatments 1–3.

**Figure 7 genes-11-00959-f007:**
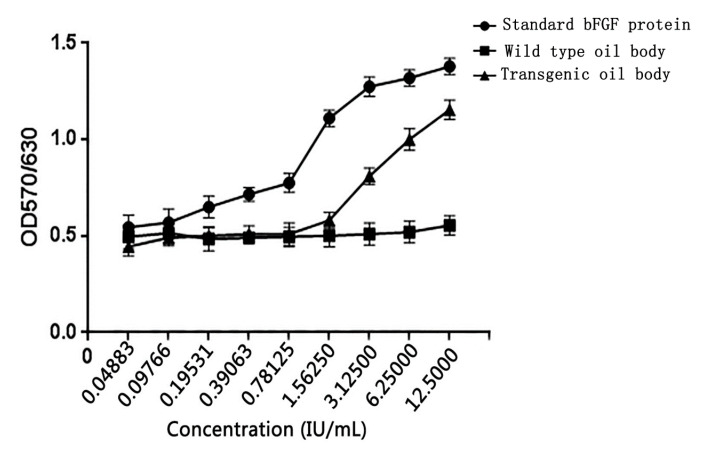
Dose–response curves for proliferation activity. Increase of NIH/3T3 fibroblast cells were expressed as the percentage increases in absorbance (570/630 nm).

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
