# Peer review of "Molecular Pharming of the Recombinant Protein hEGF-hEGF Concatenated with Oleosin Using Transgenic Arabidopsis"

_genes, 2020, doi:10.3390/genes11090959_

Round 1

Reviewer 1 Report

The work of Qiang et al., describes the successful expression of oleosin-fused hEGF in planta and demonstrates functional activity in cell lines and mice.

The subject of molecular pharming is very interesing and this work provides an alternative for mass production of the pharmaceutical protein hEGF in oil bodies.

The manuscript is descriptive, the experiments well performed and the rationale easy to follow.

I have just a few comments that, in my opinion, would help the authors and would improve the manuscript:

Section 2.1: Could the authors briefly explain (1-2 sentences) the purpose of using phaseolin promoter/terminator? It may not be that obvious to all the readers of this study.

Figure 1A: Is there a typo possibly? The correct enzyme shouldn't be BstEII instead of BstEI just before NosT?

Figure 1B & 1D: Please explain why there are double bands in the PCRs. Which primers were used to screen? Is this an expected result due to PCR setup? Maybe the lower band is the one copy of amplified hEGF and the upper band the double copy of amplified hEGF? Also in Fig. 1D, I fail to understand which four T3 transgenic lines (as stated in line 81) are positive. In fact all six of them seem identical to me. Is there any possibility of a typo and instead of four should be six (as in Fig 2) or maybe this PCR was performed in the six transgenic T2 lines (as stated in line 76)? Please clarify and provide the sequence of the primers used in this study in section 4.2.

Section 2.3: Could the authors discuss possible explanations for the observed reduction of the oil size in transgenic plants compared to control? Apart from the apparent advantage of the smaller size in skin absorption, is this result expected and often observed in in planta expression of oleosin-fused proteins?

Section 2.4, Figure 4: I personally fail to understand the purpose of this experiment. The authors used purified hEGF protein (soluble in aqueous solution, I assume) and purified oil bodies expressing oleosin-hEGF-hEGF (insoluble in aqueous solution, I assume) to monitor the degradation degree after protease treatment (in aqueous solution). Isn't it expected that the protease wouldn't degrade the oleosin-fused hEGF since the enzyme can’t access the oil bodies? Is this result truly a matter of better protein stability as stated in lines 142-143 or could it be possibly due to the "protection" of hEGF in oil bodies during the enzymatic assay? I think the authors should enrich this section. The section 4.10 is not very descriptive of how the assay was performed so possibly I misunderstood the purpose of this experiment. Please rephrase lines 294-298 so that it doesn't seem like an in-house protocol. Which "protease solution" was used for this experiment and in which final concentration (i.e. ng/mL) and final volume?

Lines 40-41: What is a small molecule polypeptide? The oil body?

Lines 50-51: "specific amino acid residue". Maybe specific size?

Line 58: "peptide-fused recombinant" --> "peptide-fused to recombinant"

Lines 169-171: "but the expression of p-EGFR after hEGF standard protein treatment was not as high as that after oleosin-hEGF-hEGF protein treatment". In Fig 6 the bands of panel A and panel B seem almost identical. Since only three (of the six) samples include beta-actin normalization, it is risky to do comparison between the two different gels. I would suggest the authors to avoid referring to higher expression of p-EGFR since the result in not so obvious. In both cases, p-EGFR is induced thus oleosin-hEGF-hEGF protein successfully activates the receptor in similar way as hEGF. This result confirms that fusion of oleosin at the N-terminus didn't alter the properties of the protein.

Line 343: please correct the cited study

Author Response

Response to Reviewer 1 Comments

The work of Qiang et al., describes the successful expression of oleosin-fused hEGF in planta and demonstrates functional activity in cell lines and mice. The subject of molecular pharming is very interesting and this work provides an alternative for mass production of the pharmaceutical protein hEGF in oil bodies. The manuscript is descriptive, the experiments well performed and the rationale easy to follow. I have just a few comments that, in my opinion, would help the authors and would improve the manuscript:

  1. Section 2.1: Could the authors briefly explain (1-2 sentences) the purpose of using phaseolin promoter/terminator? It may not be that obvious to all the readers of this study.

Thank you for your kind suggestion. We have revised our manuscript by adding/editing the following sentences. The T-DNA of the pOTB vector included a phaseolin promoter/terminator, the fusion gene, the 35S-bar gene and the nos terminator [1]. The phaseolin promoter controls tissue-specific expression during seed development. We constructed a gene expression cassette containing the fused oleosin-hEGF-hEGF genes under the control of phaseolin specific promoter (phaP)/terminator (phaT). This construct enabled the fusion gene to be specifically expressed in A. thaliana seeds. The oleosin gene was as signal peptide to carry EGF anchoring on oil body surface and express in A. thaliana seeds [2].

[1] High-efficiency production of bioactive oleosin-basic fibroblast growth factor in A. thaliana and evaluation of wound healing[J]. Gene, 2017:S0378111917307989.

[2] Expression of biologically recombinant human acidic fibroblast growth factor in Arabidopsis thaliana seeds via oleosin fusion technology[J]. Gene, 2015, 566(1):89-94.

  1. Figure 1A: Is there a typo possibly? The correct enzyme shouldn't be BstEII instead of BstEI just before NosT?

Thank you for pointing out our mistake which has been corrected in the revised manuscript. Yes, the correct enzyme is BstEII. We used the pCAMBIA1301 vector as the backbone to transform into pOTB expression vector, and the restriction site BstEII locates before the RB-T-DNA repeat of pCAMBIA1301 vector, as shown in Figure A.

Figure A: Map of pCAMBIA1301 vector

  1. Figure 1B & 1D: Please explain why there are double bands in the PCRs. Which primers were used to screen? Is this an expected result due to PCR setup? Maybe the lower band is the one copy of amplified hEGF and the upper band the double copy of amplified hEGF? Also in Fig. 1D, I fail to understand which four T3 transgenic lines (as stated in line 81) are positive. In fact all six of them seem identical to me. Is there any possibility of a typo and instead of four should be six (as in Fig 2) or maybe this PCR was performed in the six transgenic T2 lines (as stated in line 76)? Please clarify and provide the sequence of the primers used in this study in section 4.2.

Thank you for your kind suggestion. It’s supposed to be six instead of four. We have checked and revised. In the experiment, we conducted RT-PCR detection on the T3 generation of transgenic Arabidopsis thaliana seeds, and identified 6 positive lines. We screened and obtained T3 transgenic lines, and carried out RT-PCR detection of T3 transgenic seeds. The results showed that the 6 transgenic lines stably expressed the transgene.

In figure 1B & 1C, the lower band is 162 bp, which is the single hEGF band, and the upper band is 327 bp, which is the double hEGF band. In order to increase the expression of EGF, the oleosin fusion double hEGF gene was designed according to the hEGF gene sequence. The primers were designed according to the hEGF gene sequence. During the PCR reaction, the primer automatically recognized the double hEGF sequence, so there were double bands.

The hEGF-hEGF double gene sequence (327 bp)

ATGAACTCTGACTCTGAGTGCCCTCTTTCACATGACGGATACTGCTTGCACGACGGAGTGTGCATGTATATCGAGGCTTTGGACAAGTACGCTTGCAATTGTGTGGTGGGATACATCGGAGAGAGGTGCCAGTATAGGGACTTGAAGTGGTGGGAGTTGAGGATGAACTCTGACTCTGAGTGCCCTCTTTCACATGACGGATACTGCTTGCACGACGGAGTGTGCATGTATATCGAGGCTTTGGACAAGTACGCTTGCAATTGTGTGGTGGGATACATCGGAGAGAGGTGCCAGTATAGGGACTTGAAGTGGTGGGAGTTGAGGTGA

The primers sequence of hEGF:

hEGF-F: ATGAACTCTGACTCTGAGTGCCCT

hEGF-R: TCACCTCAATTCCCACCACTTC

  1. Section 2.3: Could the authors discuss possible explanations for the observed reduction of the oil size in transgenic plants compared to control? Apart from the apparent advantage of the smaller size in skin absorption, is this result expected and often observed in in planta expression of oleosin-fused proteins?

Thank you. Oleosin is a protein embedded on the surface of plant seed oil bodies [1]. The hEGF gene fused with oleosin is expressed on the surface of oil body. Experiments have also verified that oleosin is anchored on surface of oil body [2]. With the expression of oleosin in seed oil body, the surface area of the oil body decreased, thereby reducing the particle size of the transgenic seed oil body and making the oil body more stable [2-3].

[1] Frandsen G I, Mundy J, Tzen J T C. Oil bodies and their associated proteins, oleosin and caleosin[J]. Physiologia Plantarum, 2010, 112(3):301-307.

[2] Lu Y , Chi M , Li L , et al. Genome-Wide Identification, Expression Profiling, and Functional Validation of Oleosin Gene Family in Carthamus tinctorius L.[J]. Frontiers in Plant Science, 2018, 9.

[3] Siloto, Rodrigo, M,ç­‰. The Accumulation of Oleosins Determines the Size of Seed Oilbodies in Arabidopsis.[J]. Plant Cell, 2006.

  1. Section 2.4, Figure 4: I personally fail to understand the purpose of this experiment. The authors used purified hEGF protein (soluble in aqueous solution, I assume) and purified oil bodies expressing oleosin-hEGF-hEGF (insoluble in aqueous solution, I assume) to monitor the degradation degree after protease treatment (in aqueous solution). Isn't it expected that the protease wouldn't degrade the oleosin-fused hEGF since the enzyme can’t access the oil bodies? Is this result truly a matter of better protein stability as stated in lines 142-143 or could it be possibly due to the "protection" of hEGF in oil bodies during the enzymatic assay? I think the authors should enrich this section. The section 4.10 is not very descriptive of how the assay was performed so possibly I misunderstood the purpose of this experiment. Please rephrase lines 294-298 so that it doesn't seem like an in-house protocol. Which "protease solution" was used for this experiment and in which final concentration (i.e. ng/mL) and final volume?

Thank your kind suggestion. We have revised accordingly. The purpose of this experiment was to evaluate the protective effect of oil body structure on hEGF. Because of the special structure of oil body, the protease wouldn't degrade the oleosin-fused hEGF since the enzyme can’t access the oil bodies. Therefore, the oil body can stably protect hEGF.  The author have rephrased lines 294-298.  We change “The oil body system carries several advantages such as, the exogenous proteins are implanted in the oil body surface, making it easier to obtain a pure product. In addition, the oil body floats in the centrifuge to achieve the separation of the oil body from other cellular components, therefore reduces cost and time.” to “The oil body system carries several advantages such as, the exogenous proteins are implanted in the oil body surface, there is no need to purify the target protein, making it easier to obtain a pure product, and the oil body is directly applied to the skin surface. Therefore, the oil body expression system represents an ideal system for the production of therapeutic proteins”. The method is as follows: take 10 μg each of oleosin-hEGF and hEGF protein in 25 mM ammonium bicarbonate solution, add 10μL 0.25% trypsin to 100μL solution and incubate at 37 ℃.

  1. Lines 40-41: What is a small molecule polypeptide? The oil body?

Thank you. Small molecule active peptide is a kind of biochemical substance between amino acid and protein, composed of 2 to 7 amino acids [1].

[1] Jian He, Shu Wenliu, Dong Guolin, Small molecule peptide with antibacterial antiviral activity and active modifier of small molecule peptide:, 2014.

  1. Lines 50-51: "specific amino acid residue". Maybe specific size?

Thank you. Specific amino acid residue is not specific size. It refers to N-terminal or C-terminal amino acid residues. Oleosin can connect with the amino acid residues of foreign protein through its N-terminal or C-terminal amino acid residues.

  1. Line 58: "peptide-fused recombinant" --> "peptide-fused to recombinant"

Thank you for your kind suggestion. We have revised.

  1. Lines 169-171: "but the expression of p-EGFR after hEGF standard protein treatment was not as high as that after oleosin-hEGF-hEGF protein treatment". In Fig 6 the bands of panel A and panel B seem almost identical. Since only three (of the six) samples include beta-actin normalization, it is risky to do comparison between the two different gels. I would suggest the authors to avoid referring to higher expression of p-EGFR since the result in not so obvious. In both cases, p-EGFR is induced thus oleosin-hEGF-hEGF protein successfully activates the receptor in similar way as hEGF. This result confirms that fusion of oleosin at the N-terminus didn't alter the properties of the protein.

Thank you. As per your instructions, we repeated this experiment to ensure that each sample was standardized with beta-actin, and the same set of comparisons were performed in the same gel. The results showed that oleosin-hEGF-hEGF successfully activated the receptor in a similar way to hEGF. This result confirmed that the fusion of oleosin at the N-terminus didn't alter the hEGF activity.

Figure 6.  Analysis of activation of p-EGFR. (A) Activation of p-EGFR after oleosin-hEGF-hEGF treatment; (B) Activation of p-EGFR after hEGF standard protein treatment; (C) Activation of p-EGFR after normal saline treatment; (A) Activation of p-EGFR after wild-type oil body treatment.

  1. Line 343: please correct the cited study

The author has checked and revised.  I have corrected “Jahovic, N.; Güzel, E.; Arbak, S.; YeÄŸen, B.Ç. The healing-promoting effect of saliva on skin burn is mediated by epidermal growth factor (EGF): role of the neutrophils. Burns 2004, 30, 531–538.” to “Mroczkowski, B.; Ball, R. Epidermal Growth Factor: Biology and Properties of its Gene and Protein Precursor BT  - Growth Factors, Differentiation Factors, and Cytokines. In; Habenicht, A., Ed.; Springer Berlin Heidelberg: Berlin, Heidelberg, 1990; pp. 18–30 ISBN 978-3-642-74856-1.

Reviewer 2 Report

In this Manuscript, the authors report on the production of Arabidopsis Oleosin fused double hEGF recombinant protein and its properties. They have tested the oilbody (OB) produced fusion protein’s proliferation and transdermal absorption compared to the WT protein. The authors also claim that the oil body expressed oleosin-hEGF/hEGF is more skin permeable and has potential to have implications in the field of medicine. The experimental sections should be greatly improved, adding additional controls and providing additional data etc. Some of the data can be either removed totally or relegated to inclusion in the Supplementary Information (SI). Please see my detailed questions/suggestions below.

  1. In the abstract authors written RT-qPCR which has to be corrected as RT-PCR.
  2. The Arabidopsis genome contains at least 6 known oleosin genes, the authors should mention in the materials section which oleosin they have fused to the hEGF (I suppose it’s OLE1, but I urge the authors to check with Gen Bank accession number X62353.1).
  3. Figure 1 should certainly be improved. Their primer positions to be denoted in fig1A. There is no explanation on the reasons to get double bands in fig1B and D (an extra band around the size of 200bp).
  4. In my opinion fig1C and their sub pictures (a-k) is not conveying anything other than the methodology to obtain transgenic Arabidopsis expressing Ole-hEGF, hence they could be either completely removed or moved to the supplementary.
  5. The authors claim to have obtained homozygous T3 plants, but I could see some Bar sensitive yellow plants in their T3 Fig1C (h). Hence, I would expect them to provide with genomic-PCR confirming the homozygous status of their T3 plants or tone down their claim.
  6. Overexpressing OLE produces more OB, not only in seeds but also in leaves. The authors have not produced any picture showing the OB in their transgenic plants.
  7. Fig 2B western shows a faint coloration which is not very convincing, I assume they have produced the best picture they could. It would be better to use the commercially available Arabidopsis Oleosin 1 antibody if the EGF ab is not working properly.
  8. Quantifying recombinant protein by a software using the WB is not very convincing (fig 2C), at least to me. If they still want to use this method, they should provide a WB where they have loaded a series of concentration (let’s say 0-10ug of the Transgenic number 5 OB protein) and prove the quantification correlation between loaded concentration vs recombinant protein content.
  9. Figure 3A: Need explanation of sample number used in each biological replicate in the methods section (how many OBs were analyzed in each replicated). Statistics to be provided.
  10. Fig 5: Does the permeability in anyway quantifiable? They should also produce more pictures of EGF vs transgenic oil body with DAB staining (15, 30 and 45 mins) in the supplementary (at least a second biological replicate is a must). Moreover, I also would like to see the positive and negative control groups with DAB staining.
  11. Fig 6: It is very hard to see the EGFR activation at 15 min as there is no 0 min time point included. Moreover the claimed increase in EGFR activation is not convincing in ole-hEGF. I wonder why the positive and negative control groups were not included (from saline group and WT oil body group mice) in this assay as well. Which group does the B-actin immune-hybridization belongs to?
  12. Sentence starting in line 167-171 should be rewritten. Sentence starting in line 294-298 is in a different voice structure than others.

Author Response

Response to Reviewer 2 Comments

In this Manuscript, the authors report on the production of Arabidopsis Oleosin fused double hEGF recombinant protein and its properties. They have tested the oilbody (OB) produced fusion protein’s proliferation and transdermal absorption compared to the WT protein. The authors also claim that the oil body expressed oleosin-hEGF/hEGF is more skin permeable and has potential to have implications in the field of medicine. The experimental sections should be greatly improved, adding additional controls and providing additional data etc. Some of the data can be either removed totally or relegated to inclusion in the Supplementary Information (SI). Please see my detailed questions/suggestions below.

  1. In the abstract authors written RT-qPCR which has to be corrected as RT-PCR.

Thank you for your kind suggestion. We have corrected it throughout the revised manuscript.

  1. The Arabidopsis genome contains at least 6 known oleosin genes, the authors should mention in the materials section which oleosin they have fused to the hEGF (I suppose it’s OLE1, but I urge the authors to check with Gen Bank accession number X62353.1).

Thank you. While searching in Genebank for accession number X62353.1, the gene sequence of oleosin1 appeared. Its Locus tag is AT4G25140, the sequence is as follows: ATATACACATCTTTTTGATCAATCTCTCATTCAAAATCTCATTCTCTCTAGTAAACAAGAACAAAAAAATGGCGGATACAGCTAGAGGAACCCATCACGATATCATCGGCAGAGACCAGTACCCGATGATGGGCCGAGACCGAGACCAGTACCAGATGTCCGGACGAGGATCTGACTACTCCAAGTCTAGGCAGATTGCTAAAGCTGCAACTGCTGTCACAGCTGGTGGTTCCCTCCTTGTTCTCTCCAGCCTTACCCTTGTTGGAACTGTCATAGCTTTGACTGTTGCAACACCTCTGCTCGTTATCTTCAGCCCAATCCTTGTCCCGGCTCTCATCACAGTTGCACTCCTCATCACCGGTTTTCTTTCCTCTGGAGGGTTTGGCATTGCCGCTATAACCGTTTTCTCTTGGATTTACAAGTAAGCACACATTTATCATCTTACTTCATAATTTTGTGCAATATGTGCATGCATGTGTTGAGCCAGTAGCTTTGGATCAATTTTTTTGGTCGAATAACAAATGTAACAATAAGAAATTGCAAATTCTAGGGAACATTTGGTTAACTAAATACGAAATTTGACCTAGCTAGCTTGAATGTGTCTGTGTATATCATCTATATAGGTAAAATGCTTGGTATGATACCTATTGATTGTGAATAGGTACGCAACGGGAGAGCACCCACAGGGATCAGACAAGTTGGACAGTGCAAGGATGAAGTTGGGAAGCAAAGCTCAGGATCTGAAAGACAGAGCTCAGTACTACGGACAGCAACATACTGGTGGGGAACATGACCGTGACCGTACTCGTGGTGGCCAGCACACTACTAAGTTACCCCACTGATGTCATCGTCATAGTCCAATAACTCCAATGTCGGGGAGTTAGTTTATGAGGAATAAAGTGTTTAGAATTTGATCAGGGGGAGATAATAAAAGCCGAGTTTGAATCTTTTTGTTATAAGTAATGTTTATGTGTGTTTCTATATGTTGTCAAATGGTACCATGTTTTTTTTCCTCTCTTTTTGTAACTTGCAAGTGTTGTGTTGTACTTTATTTGGCTTCTTTGTAAGTTGGTAACGGTGGTCTATATATGGAAAAGGTCTTGTTTTGTTAAACTTATGTTAGTTAACTGGATTCGTCTTTAACCAC.

  1. Figure 1 should certainly be improved. Their primer positions to be denoted in fig1A. There is no explanation on the reasons to get double bands in fig1B and D (an extra band around the size of 200bp).

figure 1B and D, the lower band is 162bp, which is the single hEGF band, and the upper band is 327 bp, which is the double hEGF band. In order to increase the expression of EGF, the oleosin fusion double hEGF gene was designed according to the hEGF gene sequence. The primers were designed according to the hEGF gene sequence. During the PCR reaction, the primer automatically recognized the double hEGF sequence, so there were double bands.

The hEGF-hEGF double  gene sequence(327bp): ATGAACTCTGACTCTGAGTGCCCTCTTTCACATGACGGATACTGCTTGCACGACGGAGTGTGCATGTATATCGAGGCTTTGGACAAGTACGCTTGCAATTGTGTGGTGGGATACATCGGAGAGAGGTGCCAGTATAGGGACTTGAAGTGGTGGGAGTTGAGGATGAACTCTGACTCTGAGTGCCCTCTTTCACATGACGGATACTGCTTGCACGACGGAGTGTGCATGTATATCGAGGCTTTGGACAAGTACGCTTGCAATTGTGTGGTGGGATACATCGGAGAGAGGTGCCAGTATAGGGACTTGAAGTGGTGGGAGTTGAGGTGA

The primers sequence of hEGF:

hEGF-F: ATGAACTCTGACTCTGAGTGCCCT

hEGF-R: TCACCTCAATTCCCACCACTTC

  1. In my opinion fig1C and their sub pictures (a-k) is not conveying anything other than the methodology to obtain transgenic Arabidopsis expressing Ole-hEGF, hence they could be either completely removed or moved to the supplementary.

Thank you for your kind suggestion. We think it is very reasonable, and thus removed the Arabidopsis transformation part in Figure 1.

Figure1. Construction and detection of pOTB-hEGF-hEGF. (A) The schematic diagram of T-DNA region from pOTB-hEGF-hEGF. (B) PCR detection of recombinant plasmid. (C) T3 transgenic Arabidopsis seeds were detected by RT-PCR; M. DNA2000 Marker;1: negative control(wild type); 2-7: T3 transgenic seeds; 8: negative control(water ); 9: positive control (pOTB-hEGF-hEGF plasmid).

  1. The authors claim to have obtained homozygous T3 plants, but I could see some Bar sensitive yellow plants in their T3 Fig1C (h). Hence, I would expect them to provide with genomic-PCR confirming the homozygous status of their T3 plants or tone down their claim.

Thank you. The T3 plants were obtained after dual screening (BASTA and RT-PCR) of their parent T1 and T2 generations, thus they were stable homozygous transgenic lines.

  1. Overexpressing OLE produces more OB, not only in seeds but also in leaves. The authors have not produced any picture showing the OB in their transgenic plants.

The T-DNA of the pOTB vector included a phaseolin promoter/terminator, the fusion gene, the 35S-Bar gene and the nos terminator. The phaseolin promoter controls tissue-specific expression during seed development. We constructed a gene expression cassette containing the oleosin-hEGF-hEGF gene under the control of the phaseolin specific promoter (phaP)/terminator (phaT). This construct enabled the fusion gene to be specifically expressed in A. thaliana seeds. The oleosin gene was as signal peptide to carry EGF anchoring on oil body surface and express in A. thaliana seeds [2]. Therefore, oleosin-hEGF was only expressed in oil body of transgenic seeds.

 [1] High-efficiency production of bioactive oleosin-basic fibroblast growth factor in A. thaliana and evaluation of wound healing[J]. Gene, 2017:S0378111917307989.

[2] Expression of biologically recombinant human acidic fibroblast growth factor in Arabidopsis thaliana seeds via oleosin fusion technology[J]. Gene, 2015, 566(1):89-94.

  1. Fig 2B western shows a faint coloration which is not very convincing, I assume they have produced the best picture they could. It would be better to use the commercially available Arabidopsis Oleosin 1 antibody if the EGF ab is not working properly.

Thank you for your comments. At present, there is no oleosin1 antibody commercially available, and if the oleosin1 antibody is used for immunohybridization, the wild-type oleosin1 will also hybridize with it, not easy to distinguish. So we chose EGF antibody to hybridize the fusion protein. Even repeating the experiment, the results were consistent, proving that the EGF antibody is very effective and can work properly.

  1. Quantifying recombinant protein by a software using the WB is not very convincing (fig 2C), at least to me. If they still want to use this method, they should provide a WB where they have loaded a series of concentration (let’s say 0-10ug of the Transgenic number 5 OB protein) and prove the quantification correlation between loaded concentration vs recombinant protein content.

Thank you. Yes, The WB quantitative method belongs to semi-quantitative method. We quantified the target protein by drawing the standard curve of the standard protein. EGF standard protein of different concentrations was detected by western blot, and the gray value of each band was analyzed with Quantity One software. The standard curve is drawn and the X axis shows the EGF standard protein of different concentrations, and the Y axis shows the gray value. At EGF standard protein 1000 ng, the gray value equals 7220, the regression equation is: y=808.57x+2120, R2=0.9902, and finally the expression level of the target protein is analyzed according to the gray value of the target protein band, such as the highest expression level of target protein was calculated 14.83 ng/μL oil body (Supplementary documents, Table 1).

Figure 2. Protein expression from T3 transgenic Arabidopsis. (A) Western blot; (B) greyscale values and (C) the standard curve of the grayscale values from various concentrations of hEGF standard protein; (D) SDS-PAGE of oil body; (E) western blot of fusion protein and (F) quantitation of fusion protein from T3 Arabidopsis seeds. M: Protein marker, 1-6: oil body in T3 Arabidopsis; WT: oil bodies from WT Arabidopsis.

  1. Figure 3A: Need explanation of sample number used in each biological replicate in the methods section (how many OBs were analyzed in each replicated). Statistics to be provided.

Thank you for your suggestion. We have revised. The samples of each experiment were carried out in 3 biological replicates. The data of repeated experiments were expressed as the mean and standard deviation (mean ± SD). We used analysis of variance (ANOVA) method for statistical analysis. If the p value was <0.05, the difference was considered significant.

  1. Fig 5: Does the permeability in anyway quantifiable? They should also produce more pictures of EGF vs transgenic oil body with DAB staining (15, 30 and 45 mins) in the supplementary (at least a second biological replicate is a must). Moreover, I also would like to see the positive and negative control groups with DAB staining.

Thank you. After hybridization with hEGF antibody and DAB staining, it was observed that the skin infiltrated with hEGF was brown. Permeability could be analyzed by counting the area of the brown areas, but it is only a rough analysis, so there is no quantitative way, but only through observation to judge the permeability at different times. The experiment was repeated twice with biological replicates and pictures of another biological repeat was set up. The specifications are shown in the supplementary file (Supplementary documents, Figure S1). During this experiment, EGF protein is the positive group and normal saline is the negative control group. The absorption capacity of oil body expressed oleosin-hEGF-hEGF was better than hEGF protein. Compared with the hEGF protein, the proportion of brown stained areas for transgenic oil body was larger. In addition, the staining intensity of transgenic oil body was greater than that of hEGF protein and normal saline treatment group.

Reviewer 3 Report

Dear authors,

I carefully read your paper and I think it has enough quality to publish with Genes, MDPI. However, there are some minor comments I would like to know your opinions about before getting consider if could be accepted from the respected editorial team.

  • Your paper inserted a complete power of concentration on the biopharming of a recombinant protein with potential clinical applications under the plant expression system. Can such strategies cause allergic reactions after the medicinal administration of such products? Is there any strategy to reduce the potential safety issues for improving the large-scale application of such GMO products?
  • Although plant expressing systems are a cost-effective and suited platform to produce a variety of recombinant proteins with applications for both animal and human systems; however, there are some key factors that limited the extensive use of these systems for producing specific types of recombinant proteins. The author should provide some statements about the demerits of plant systems for the production of such proteins in their conclusion or discussion sections.
  • Non-human glycosylation is a major shortcoming of producing recombinant proteins. Is there any possibility the introduced system in this paper meets up such biochemical processes?
  • In the case of hEGF protein, correct folding, solubility and the ratio of its permeability may decrease the quality of this recombinant protein whenever it expressed under plant expression systems. Nonmammalian cells may have difficulty producing the correct folding of human proteins; therefore, these risks must be addressed with purification procedures, adding to the cost of downstream processing.
  • Figure 1 hasn’t enough quality. Please provide the original size of applied sections in this figure without pixel changing in the supplementary file.

Author Response

Dear authors, 

I carefully read your paper and I think it has enough quality to publish with Genes, MDPI. However, there are some minor comments I would like to know your opinions about before getting consider if could be accepted from the respected editorial team.

  1. Your paper inserted a complete power of concentration on the biopharming of a recombinant protein with potential clinical applications under the plant expression system. Can such strategies cause allergic reactions after the medicinal administration of such products? Is there any strategy to reduce the potential safety issues for improving the large-scale application of such GMO products?

Thank you for your kind comment.  Our team has completed the experiment on allergic reactions. From the current experimental results, no adverse reaction has occurred, and the target protein is expressed in plants, so we think it is safe and non-irritating. In addition, there are also some differences between transgenic plants and plant bioreactors. The exogenous protein expressed in plant oil body system is not used for edible purpose, but for medical use by extracting oil body. It could be used for external use, and does not involve oral administration. The expression exogenous protein in plant oil body system is based on the structure of oil body to carry hEGF without purification of target protein. It can be directly extracted from oil body for application, which can reduce the purification process and as well as cost. Therefore, we believe that it is safe, reliable and easy to mass produce.

  1. Although plant expressing systems are a cost-effective and suited platform to produce a variety of recombinant proteins with applications for both animal and human systems; however, there are some key factors that limited the extensive use of these systems for producing specific types of recombinant proteins. The author should provide some statements about the demerits of plant systems for the production of such proteins in their conclusion or discussion sections.

Thank you for your kind suggestion. We have revised accordingly. Plant bioreactors have the irreplaceable advantages of animal and microbial bioreactors: the homozygotes obtained after selfing of genetically modified plants can ensure the stable inheritance of new traits. Similarly, the expression product can undergo post-translational processing and protein glycosylation in plant cells, making the three-dimensional structure more natural, and the recombinant protein and the natural protein have nearly the same immunogenicity and biological activity. Plant expression systems produce medicinal substances with low cost, wide sources, and easy mass production. Compared with animal and microbial reaction systems, plants do not contain pathogens that harm human health. However, the ratio of exogenous recombinant proteins to the total plant biomass is usually very low, which increases the cost of separation and purification. Therefore, how to increase the expression of exogenous proteins in plants, reduce downstream production costs, and simplify the purification process is a problem in plant reactor research. However, the use of plant seeds to express foreign proteins has attracted much attention, mainly because of the high protein content of plant seeds. During the seed maturation process, about 95% of the water is actively lost, the hydrolytic enzyme activity in the cell is greatly reduced, and the recombinant protein can be stored in the seeds for a long time without being degraded. The oleosin in the seed has the characteristics of both hydrophilic and lipophilic. It is embedded in the surface of the oil body in a membrane embedding manner in the plant seed. Oil body protein is expressed at a high level in oil crop seeds and is easily separated. At present, various proteins such as hirudin and FGFs have been expressed in the oil body bioreactor, and the expression level can reach commercial standards.

  1. Non-human glycosylation is a major shortcoming of producing recombinant proteins. Is there any possibility the introduced system in this paper meets up such biochemical processes?

Thank you. Each expression system has its own advantages. The advantage of plant expression system is that plants can fold polypeptides and integrate subunits efficiently. In the process of processing the target protein, in order to be able to glycosylate the protein in plants, glycosylation units can be added [1]. Therefore, the plant expressing foreign protein system may satisfy the biochemical process of protein glycosylation. At present, the E. coli system is widely used and can express many proteins. However, the E. coli system is prokaryotic expression and has no post-translational processing and modification.

  1. In the case of hEGF protein, correct folding, solubility and the ratio of its permeability may decrease the quality of this recombinant protein whenever it expressed under plant expression systems. Nonmammalian cells may have difficulty producing the correct folding of human proteins; therefore, these risks must be addressed with purification procedures, adding to the cost of downstream processing.

Thank you for your kind suggestion. We have revised accordingly. It is difficult for non-mammalian cells to fold the protein correctly, because in the process of protein formation, the co-expression of molecular chaperones is a promising way to improve protein solubility and folding, but it is protein-specific. Even in the presence of molecular chaperones, lack of disulfide bonds or no post-translational modifications cannot make it fold normally. However, non-mammalian cells such as E. coli lack the co-expression of molecular chaperones and the formation of disulfide bonds [1]. But the plant bioreactor provides a eukaryotic modification site for the expression of foreign proteins to ensure the correct post-transcriptional and post-translational modification of the expressed protein [2].

[1] Hannig G , Makrides S C . Strategies for optimizing heterologous protein expression in Escherichia coli.[J]. Trends in Biotechnology, 1998, 16:54.

[2] Hein M B , Hiatt A . Transgenic plants expressing assembled secretory antibodies[J]. 2002.

  1. Figure 1 hasn’t enough quality. Please provide the original size of applied sections in this figure without pixel changing in the supplementary file.

Thank you. We have improved figure 1 in the revised manuscript.

Round 2

Reviewer 2 Report

I would like to thank the Authors for considerably improving their manuscripts. My questions were all addressed, and I am convinced; however, I still feel to reply to some of their comments and ask them to address in their revised version,

  1. Their explanation on their T3 plants are bit foggy. They claim to have done RTPCR, segregation analysis with Basta (3:1 in T2), yet find some segregation in T3 (Fig1C(h) previous version), which should not happen unless the pool still contains heterozygous. Even though this figure is modified now, the fact remains the same. So, I highly recommend just to use T3 line and not homozygous T3 lines.
  2. I appreciate them for the statistical testing, but it is still not shown in fig3A, please include the Asterisk/s in the figure according to the significance.
  3. Experimental materials (2.1) still doesn’t mention the gene is OLE1 or tag their ATG number.

Small correction for their answer on OLE1ab: I agree with their explanation on not using the OLE1 antibody to detect the recombinant protein. However just for their information, the Oleosin ab is commercially available for a long time now and the size differences (native 18.5kD vs recombinant 31kD) could be easily seen in a western blot.

http://www.phytoab.com/ole1-anti-oleosin-18-5-kda-antibody

Author Response

Response to Reviewer 2 Comments

I would like to thank the Authors for considerably improving their manuscript. My questions were all addressed, and I am convinced; however, I still feel to reply to some of their comments and ask them to address in their revised version.

  1. Their explanation on their T3 plants are bit foggy. They claim to have done RTPCR, segregation analysis with Basta (3:1 in T2), yet find some segregation in T3 (Fig1C (h) previous version), which should not happen unless the pool still contains heterozygous. Even though this figure is modified now, the fact remains the same. So, I highly recommend just to use T3 line and not homozygous T3 lines.

Thank you for your kind suggestion. As per your instructions, we will now use just T3 lines in our experiments. We have edited the text in revised manuscript. Sorry, our description in the original text was not clear, which caused misunderstandings. We first carried out Basta screening on the T1 generation transgenic seeds. Most of the plants turned yellow and died (the previous version of Fig1C (h)), and the survived plants were transferred to a new pot for continued cultivation. The T2 generation transgenic seeds were harvested, and then the T2 generation seeds were continued to be screened and expanded (the segregation ratio conformed to Mendelian inheritance rules) to obtain the T3 line, which was used for further experimentation.

In the previous version of Fig1C (h), the samples we tested were T2 transgenic seeds, so there was segregation. In the later test, we replaced them with T3 transgenic seeds. The other experiments in the later period used T3 generation transgenic seeds. We also modified the respective figure accordingly.

  1. I appreciate them for the statistical testing, but it is still not shown in fig3A, please include the Asterisk/s in the figure according to the significance.

Thank you for correction. In the revised manuscript, we added Asterisks in figure 3A, according to significance.

Figure 3. Measurement and observation of size and microstructure from transgenic oil body and normal oil body. (A) Particle size. (B) Microstructure observation. a. dyed normal oil bodies via nile red. b. dyed transgenic oil bodies via nile red. The images were shown at 40×10 magnification (Scale bars=10µm), **P <0 .01 vs. normal group.

  1. Experimental materials (2.1) still doesn’t mention the gene is OLE1 or tag their ATG number.

Thank you for kind reminder. We added the gene name (OLE1) in (subsection 2.1. of) Materials and Methods.

Small correction for their answer on OLE1ab: I agree with their explanation on not using the OLE1 antibody to detect the recombinant protein. However just for their information, the Oleosin ab is commercially available for a long time now and the size differences (native 18.5kD vs recombinant 31kD) could be easily seen in a western blot.

http://www.phytoab.com/ole1-anti-oleosin-18-5-kda-antibody

Thank you very much for your kind suggestion. We have not purchased oleosin1 antibody before which caused trouble for our research group. We really need it in our experiment. We can further test the genetically modified strains. Thank you very much for the URL, you provided to purchase antibody.
